# A Multiaxis Tool Path Generation Approach for Thin Wall Structures Made with WAAM

**Matthieu Rauch** [1,2,*] **, Jean-Yves Hascoet** [1,2] **and Vincent Querard** [1,2]

1    GeM-UMR CNRS 6183, Centrale Nantes, 1 Rue de la Noe, 44321 Nantes, France; jean-yves.hascoet@ec-nantes.fr (J.-Y.H.); vincent.querard@ens-rennes.fr (V.Q.)
2    Joint Laboratory of Marine Technology (JLMT), Centrale Nantes-Naval Group, 44321 Nantes, France
*    Correspondence: matthieu.rauch@ec-nantes.fr

**Abstract:** Wire Arc Additive Manufacturing (WAAM) has emerged over the last decade and is dedicated to the realization of high-dimensional parts in various metallic materials. The usual process implementation consists in associating a high-performance welding generator as heat source, a NC controlled 6 or 8 degrees (for example) of freedom robotic arm as motion system and welding wire as feedstock. WAAM toolpath generation methods, although process specific, can be based on similar approaches developed for other processes, such as machining, to integrate the process data into a consistent technical data environment. This paper proposes a generic multiaxis tool path generation approach for thin wall structures made with WAAM. At first, the current technological and scientific challenges associated to CAD/CAM/CNC data chains for WAAM applications are introduced. The focus is on process planning aspects such as non-planar non-parallel slicing approaches and part orientation into the working space, and these are integrated in the proposed method. The interest of variable torch orientation control for complex shapes is proposed, and then, a new intersection crossing tool path method based on Design For Additive Manufacturing considerations is detailed. Eventually, two industrial use cases are introduced to highlight the interest of this approach for realizing large components.

**Keywords:** additive manufacturing; WAAM; AM tool paths generation

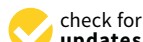

## 1. Introduction

The rise of Additive Manufacturing (AM) over the last twenty years offers new opportunities such as cost reduction and freedom of manufacturing, depending on the type of component. Among the AM processes referenced by ISO/ASTM 52900 [1], Direct Energy Deposition (DED) processes build the parts layer by layer and feed the material closed to a CNC controlled heat source effector: Laser Metal Deposition is more dedicated to challenging geometries [2], functional graded materials parts [3], or repair [4] whereas Wire Arc Additive Manufacturing (WAAM) is more suitable for building large structural parts [5]. WAAM provides high deposition rate, cost competitiveness of the equipment, large build envelops [6] etc. It has consequently shown its efficiency for various high-dimensional parts in various materials like steel, aluminum, and titanium [7,8]. Most of the existing challenges associated to the WAAM process are reviewed by Jafari et al. in their review paper [9] and by Treutler and Wesling in [10].

As any other AM system, WAAM implementation is based on three functional elements: a motion system, a heat source, and a feedstock. WAAM feedstocks are welding wires, thin metallic rods with distinct chemical composition. There are different commercially available wires (titanium alloys, steel, aluminum alloys, Inconel, etc.), and these can be found in various sizes (0.6–2.4 mm). As for welding, these wires fulfill some requirements to be suitable for WAAM, such as wire surface smoothness, uniformity, and absence of scratches.

WAAM heat sources are high performance welding generators. Several welding technologies can be employed to generate the arc. Conventional welding processes such as gas metal arc (GMAW), plasma arc (PAW), and tungsten inert gas (TIG). GMAW is the most developed, as the filler wire is fed coaxially, hence avoiding deposition process variation often experienced in PAW and TIG during rotation of the welding direction around its axis [11]. A variant of GMAW, the Cold Metal Transfer (CMT), is frequently employed due to his advantages for AM applications: low spatter, low heat input and improved deposition using dip transfer mode [7]. CMT torch motor manages the reciprocating motion of the wire: when the advancing wire meets the piece to be welded, the arc extinguishes, and the wire retracts to a predefined optimal position for the ignition process and transfer of molten metal. The WAAM motion system is usually a 6 degrees of freedom robotic arm, as its flexibility is well adapted to the process requirements, and it offers valuable possibilities such as tool axis flexible orientation and large workspace compared to its size. However, some companies have developed a WAAM solution based on machine tool structures. All types of structures employed are CNC controlled.

As the welding torch is placed at the end of the robot, WAAM process implementation consists in controlling the tool position and orientation in the working space, while using the most suitable welding parameters. WAAM tool path generation is consequently a key issue and has been investigated by many researchers [12]. Various tool path generation techniques have been developed, most of them focusing on massive sections manufacturing made by overlaying several welding beads on the same layer: several patterns have been studied and evaluated such as raster, zigzag, and spiral [13] which are derived from machining tool paths pattern [14,15].

The connection with other CNC processes tool path generation methods is quite natural to make. CAM software vendors, such as Autodesk [16] or Siemens [17], have developed their own software module for WAAM, based on their existing solutions for high speed machining, using existing tool path generation algorithms, kinematics 3D simulation of robotic processes, and post-processing modules. In parallel, for FDM machines, stand-alone software + hardware solutions dedicated to WAAM have emerged as well [18,19].

However, tool path parameters selection and control have not been discussed in detail so far. In addition, there are hardly any methodologies available to realize a thin wall component with WAAM, going from the CAD model to the real manufactured part. Thin wall structures are defined and manufactured by single bead deposition layer by layer.

This paper consequently proposes a generic multiaxis tool path generation approach for thin wall structures made with WAAM and is organized as follows. Section one introduces the different steps to follow with the associated technological and scientifical challenges. Then, two industrial use cases are detailed to highlight the interest of such approach for realizing large components. The final discussion focuses on the key aspects developed in this paper.

All the experimental results provided in the paper have been obtained by using the robotic cell of the laboratory whose size is 6 m × 6 m × 4 m (Figure 1). It contains a large size robot (reach 2830 mm) equipped with welding torches and CMT welding generators. Specific process synergies were selected to define the material dependent process parameters (wire feed speed, current, voltage, travel speed, etc.) WAAM tool paths were generated using a commercial CAD/CAM software suite for multiprocess manufacturing (HSM and AM) associated to specifically developed algorithms.

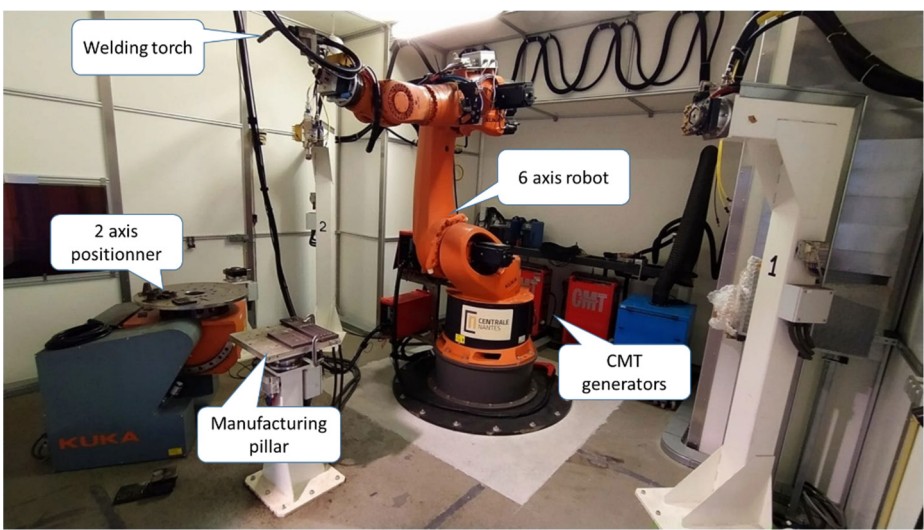

**Figure 1.** WAAM robotic cell of the laboratory.

## 2. Proposed Tool Path Generation Methodology

The method proposed here is dedicated to thin wall components realized with the WAAM process. Several key steps are detailed in this section. First, the focus is placed on the CAD/CAM/CNC numerical data chain and the associated challenges. Then, slicing and part orientation in the working space and process planning issues are discussed. The generation of torch axis-controlled tool paths is proposed as a third step. The fourth step introduces a Design for Additive Manufacturing (DFAM) approach for wall intersections and junctions manufacturing.

### 2.1. CAD/CAM/CNC Data Chain Preparation

As they emerged on a grand scale during the 2000's, the Additive Manufacturing processes were based since their early stages on Computer Aided hardware and software tools. Even though the physical phenomena are very different from machining, the numerical data chain developed for AM is based on the same organization.

This organization of CAD/CAM/CNC data is a direct legacy from the second industrial revolution with the sequencing of manufacturing tasks in the industry. Going from one of its links to another is achieved by meeting upstream and downstream requirements, which leads to standardized data exchange formats.

The first stage of the data chain, the Computer Aided Design (CAD), aims to build the numerical model of the part to manufacture. Geometries and Material properties are chosen to meet the functional, assembly, and manufacturing requirements.

The second stage is for Computer Aided Manufacturing (CAM). Compared to other CNC controlled manufacturing processes, CAM for AM needs a specific stage named "Slicing", which is carried out in coordination with the orientation of the part in the AM machine working space. The objectives are multiple: reduce the need of support material, enable the manufacturing of complex geometries, improve part accuracy, decrease manufacturing times, etc. Some key aspects of the slicing operation are given in the next section of this paper.

After the slicing operation, the generated tool paths for all the layers are integrated into the manufacturing program. The latter s run on the manufacturing equipment using the same control principles than other CNC processes. For example, simulation software tools can be used for verification and kinematic optimization purposes.

The manufacturing program is then converted to a language specific to the manufacturing equipment, and the realization of the component is carried out on the selected equipment. After the part is obtained, control and inspection operations can be carried out.

The CAD/CAM/CNC data chain organization for WAAM has to face several challenges, and most of them can benefit from the experience acquired from other CNC

processes such as machining. Hence, the use of neutral files format, such as the STEP-NC approach [20], can be of great help to benefit from high level information about the manufacturing data chain.

As shown in Figure 2, the CAM stage is the most critical and is divided into several steps that are strongly interconnected.

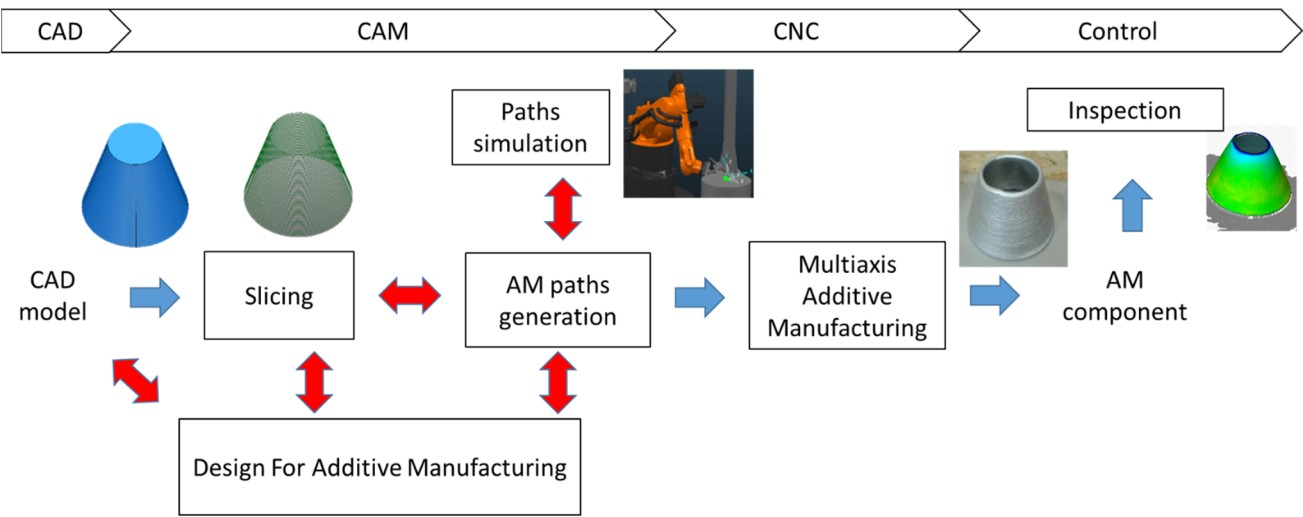

**Figure 2.** From CAD model to manufactured part with AM.

The slicing operation is performed at an early stage before generating the explicit tool paths, but it depends strongly on the manufacturing equipment characteristics and the process capability. Then, depending on the CAD model geometry and setup, the tool path generation calls for multiaxis toolpaths so that the AM torch can be oriented in the working space. Process parameters such as layer height, bead width, and energy input commands are associated to the geometry of the tool paths created to make the manufacturing program.

### 2.2. Slicing and Support Set Up Orientation

As for machining, the part orientation into the manufacturing equipment working space has to be decided at first, according to process planning and technological aspects. The deposition direction can be defined to be normal to slice surface. This direction also determines the substrate orientation to start the process.

In practice, some components show obvious manufacturing directions, such as the revolution axis for axisymmetric parts. An example of setup orientation is provided by Figure 3, where several deposition directions are considered for a single geometry.

For the cases where the deposition direction does not coincide with the favorable direction, asymmetric defects such as melt pool collapse can be induced from the gravity effects. The deposited bead is then not parallel to the slicing surface which may lead to variable manufacturing conditions. These phenomena increase with the number of layers until the process is not working at all.

As a result, a first critical aspect is to identify the favorable direction. If it cannot be easily determined, one should orient the part to minimize the overhang angle, in order to contain the effects of gravity.

When the setup orientation has been defined, the CAD model can be sliced. The slicing operation consists in dividing the numerical model into a series of layers, by intersecting its geometry with a set of slicing surfaces. On each slice, the intersection of the slicing surface with the numerical model defines the frontier curves used to generate the manufacturing tool paths: filling paths, contour paths, and support paths (if any), together with the technological process parameters (bead width and height, filling strategies, manufacturing tool orientation, etc.)

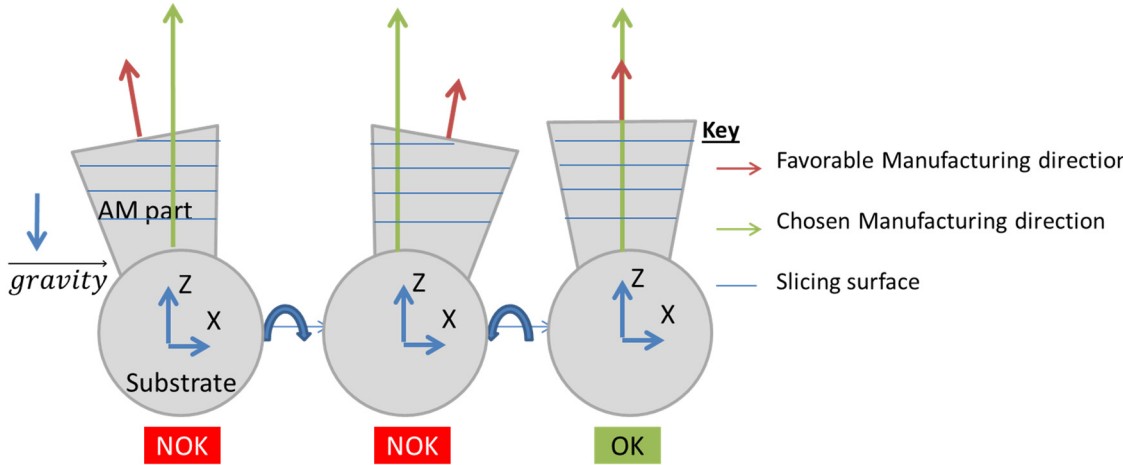

**Figure 3.** Example of manufacturing direction selection.

The common solution is to use a unidirectional slicing based upon a set of parallel plans whose distance depends on a predefined layer height. The tool paths obtained are defined in a 2D space for each layer and can be manufactured by a quite basic kinematics machine, as by most of the 3D printers available on the market. With this technology, undercuts and fragile zones are realized by the use of support material which is deposited in each layer in same way as the part material. This solution highly simplifies the tool paths generation stage but implies an additional stage to remove the support from the part. This latest stage can be very difficult to handle in practice, in particular for metallic parts.

However, for some AM processes as Wire Arc Additive Manufacturing (WAAM), it is highly valuable to use non parallel plans or parameterized surfaces as slicing surfaces [8]. As depicted in Figure 4, a slicing done by parallel plans would require support and the drawbacks associated. In contrast, a slicing along predefined angular sections or—even better—along the neutral axis leads to manufacture the whole part without any support while having accurate surfaces and homogeneous thickness. This enhanced slicing strategies call for a total control of the deposition tool axis enabled by the equipment and are usually dedicated to DED (Direct Energy Deposition) processes.

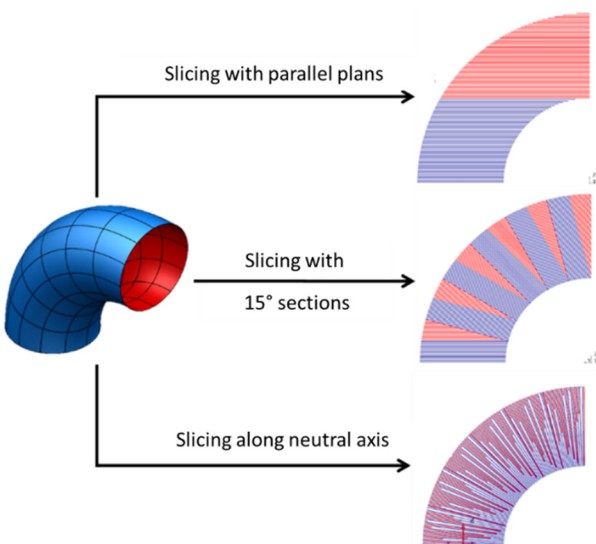

**Figure 4.** Several slicing strategies for a single geometry.

The distance between layers is set by the predefined layer height (LH) associated to the WAAM process parameters. The objective is to keep the Contact-Tip-to-Work Distance (CTWD) (distance between the welding torch and the deposited area) constant.

This parameter is essential for accurate control, and the programmed value is as close as possible to the actual layer height obtained during the process. If the programmed layer height is too high, the stick out distance increases, which makes gas protection inefficient and creates electric arc instabilities. If the programmed layer height is too small, the welding torch gets closer to the deposited material, and the gas pressure increases on the bead, which can lead to a higher porosity rate, and collision between torch and bead can happen. Moreover, if the manufacturing conditions are not constant, part homogeneity can be strongly affected. Process Monitoring such as [21] can be of great help to maintain a constant stick out distance during the process.

### 2.3. Variable Tool Direction Additive Manufacturing

The WAAM process enables the realization of complex geometries with no need to realize support geometries as well as the part. However, this advantage calls for a precise control of the torch orientation during the process, especially for undercut areas. Hence, a generic rule is to orient the torch perpendicularly to the deposition direction and the thin wall orientation as shown by Figure 5.

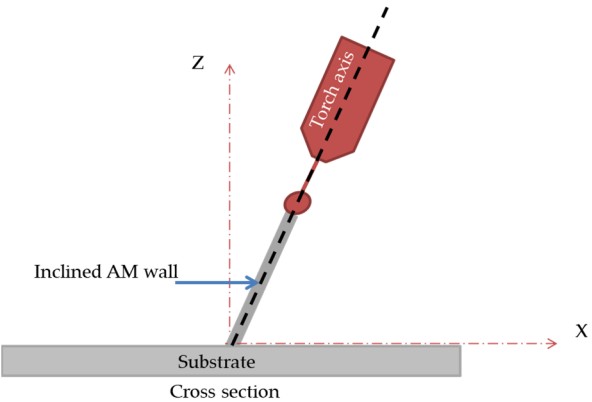

**Figure 5.** Manufacturing of an inclined thin wall by using torch axis orientation.

After having identified these tool orientation considerations, the tool path generation process can be based on approaches developed for multiaxis machining: positioning the torch angle respectively from the geometry to manufacture, defining smooth tool orientation changes, using guide curves or double splines paths for axis control, etc. Figure 6 illustrates the use of guide curves to orient the torch axis correctly along the tool path. The tool orientation angle evolves linearly between the given guides, whose number and position enable to control its evolution along the path.

However, the torch axis shall not systematically be perpendicular to the deposition direction, as it is essential to control the effects of gravity. In this way, a solution would be to use supports, such as in the three-axes FDM process implementation where supports are used to overtake overhangs and part distortion issues, but this should be avoided with DED processes such as WAAM. In contrast, using a multiaxis part positioner to carry out the deposition always in favorable conditions with regards to gravity can be a valuable technological solution [22], but it implies to use manufacturing equipment capable of part orientation and generates a tool path generation increased complexity as the articular motions have to meet the requirements associated with gravity. In addition, it is hardly possible to use variable support orientation for some large parts such as aeronautical panels.

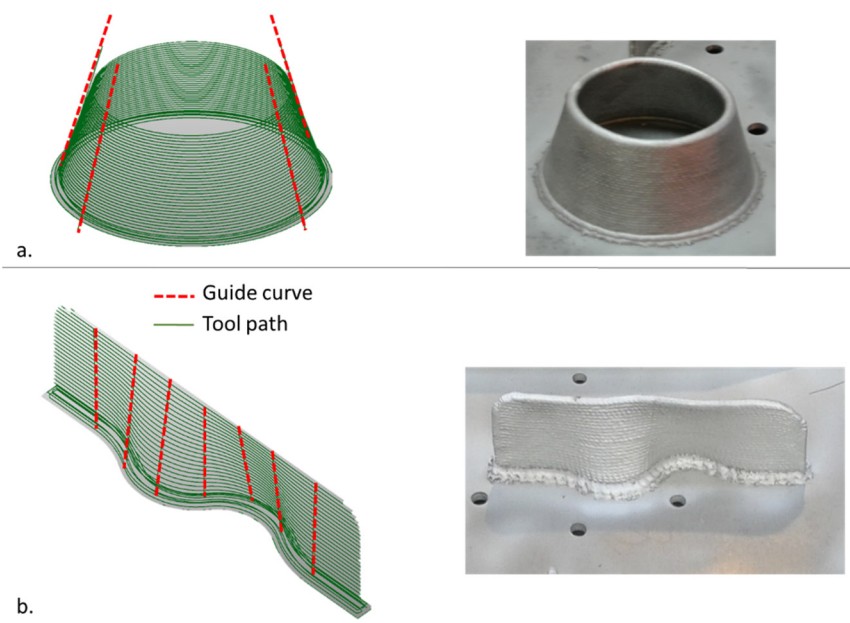

**Figure 6.** Tool path generation with guide curves to orient two axes: (**a**) 15° cone and (**b**) variable inclination wall.

An interesting technique, inspired from overhead welding, can be employed: using a predefined tool orientation from the horizontal plan (for example 20° for aluminum alloys) enables to deposit walls horizontally, without the use of any positioning device to orient the part. An example of WAAM horizontal wall is given in Figure 7.

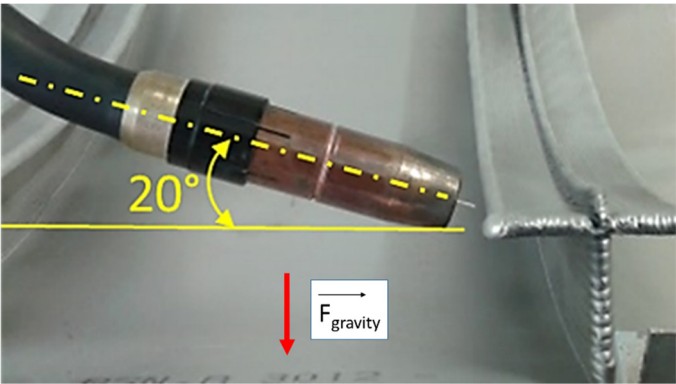

**Figure 7.** WAAM torch position to manufacture horizontal walls.

Another important aspect for the generation of multiaxis AM tool paths is the control of the tool paths starts and stop points. Electric arc start/stop should be avoided as much as possible as it generates material excess or lacks, in contrast with single continuous bead deposition, which provides homogeneous surface roughness as the manufacturing conditions are more stable. Manufacturing times are improved as well as arc start and stops require pausing the process until stable conditions are recovered. In addition, this strategy is well adapted to components that can be manufactured with closed tool paths. For components which are manufactured with open tool paths such as single walls, switching the manufacturing direction between consecutive layers compensates efficiently the geometrical errors generated by arc starts and stops.

### 2.4. Intersections and Junctions Control by DFAM Approach

Intersections and junction control is a key issue for single bead component additive manufacturing. Manufacturing a two-walls crossing may lead to an excess of material

before and a lack of material after it, when travel directions are not alternated between paths. Some strategies have been developed by researchers to cope with this situation by switching manufacturing direction between consecutive layers as shown in [23,24] (Figure 8). However, there is still a peak of material at the intersection point, as material is deposited twice at this point [25]. This defect increases with the number of layers.

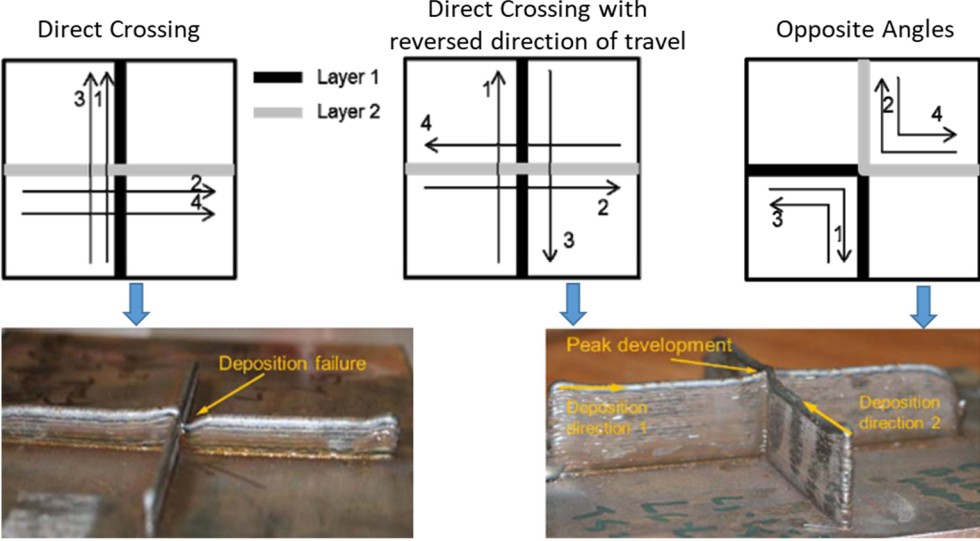

**Figure 8.** Examples of intersection crossing (adapted from [23]).

Another method consists in avoiding overlapping layers at the crossings by changing the manufacturing direction at the intersection. This solves the problem of cavity after the intersection, but there is still a double deposition at the intersection.

This situation provides several drawbacks for both process and components: the material excess quantity is linked to the component height; the WAAM torch can collide with it, and for multiple intersections (three walls and more), the defect is amplified. The material quality at the intersection is poor and creates weaknesses.

To overcome these difficulties a DFAM based approach is proposed in this paper. This method consists in modifying the local geometry of the CAD numerical model at the intersections. The objective is to avoid material excesses at these points. This approach uses the concept of "roundabouts" instead of direct crossing, as illustrated in Figure 9 with a two-walls intersection. The geometry is modified into one central cylinder with four intersecting walls. Each manufacturing layer can be interpreted as a four-tracks roundabout. This geometry has been obtained by design experience with the objective of avoiding that any point of the intersection belong to more than two intersecting walls.

As depicted on Figure 9a, the manufacturing strategy is organized as follows:

1.   Path 1 realizes walls A and B and the linking portion of the central circle;
2.   Path 2 realizes walls B and C and the linking portion;
3.   Path 3 realizes walls C and D and the linking portion;
4.   Path 4 realizes walls D and A;
5.   Path 5 realizes the central circle.

With this approach, layers are manufactured by pairs; each pair of layers needing five paths. After Paths 1 to Path 4, each of the tracks has been increased by two layers, whereas the central cylinder only by one. Path 5 is to compensate this lack of material. No crossing path is needed, arc starts and stops are limited, so the material height is consequently balanced without any peak or lack of material.

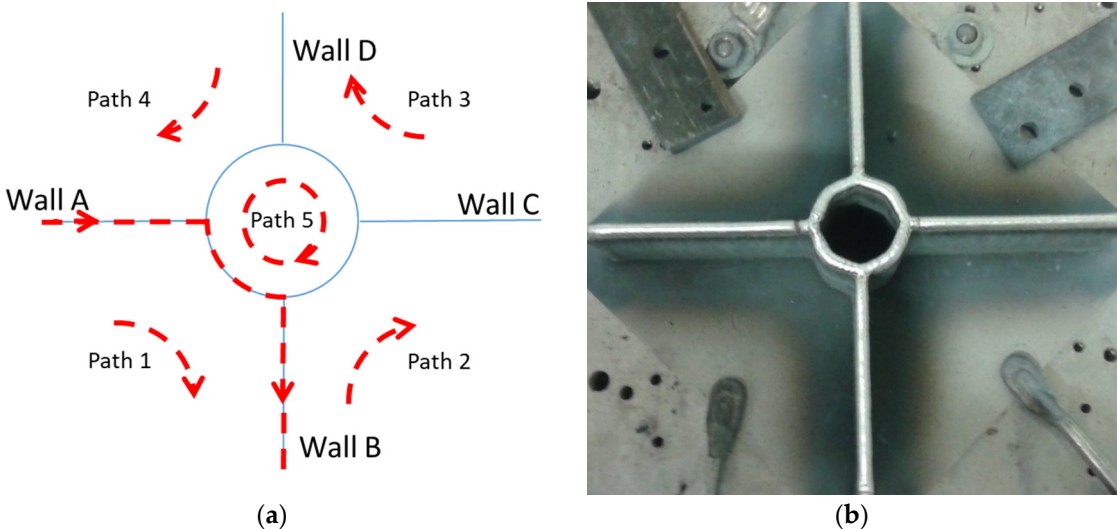

**Figure 9.** Roundabout-type intersection for two walls crossing: (**a**) manufacturing paths and (**b**) manufactured intersection.

It is also important to highlight the flexibility of the proposed approach. Depending on the functional requirements or surface finishing considerations, the central cylinder profile can be replaced by ellipsoids, polygons, arc curves, etc., and as shown in Figure 10, this DFAM approach applies to any single path walls intersections.

| Four-track intersection | Manufacturing profile | Step 1 | Step°2 | Step°3 |
| :---: | :---: | :---: | :---: | :---: |

| Three-track intersection | Manufacturing profile | Step 1 | Step°2 | Step°3 | Step°4 |
| :---: | :---: | :---: | :---: | :---: | :---: |

| Five-track intersection | Manufacturing profile | Step 1 | Step 2 | Step 3 | Step°4 |
| :---: | :---: | :---: | :---: | :---: | :---: |

**Figure 10.** Multiple track intersection crossing with the proposed method.

As a result, this DFAM approach calls for geometry adaptation and complexifies the tool path planning as layers are grouped by pairs for the deposition. However, it

provides a strong advantage by smoothing the material deposition whatever the number of intersecting walls. The modified geometry, with the central cylinder, is also more rigid and resistant.

## 3. Application on Industrial Parts

In this section, the methodology proposed in the paper is applied on two different industrial components. They have different geometries and obey to different functional requirements, but both can benefit from the approach. The first component is a cruise ship propeller. The focus is made on the DFAM and CAD/CAM/CNC data chain of the approach. The second part is a demonstrator of aeronautic structural panel which highlights the interest of DFAM controlled intersections and variable torch axis manufacturing.

### 3.1. Cruise Ship Propeller

The first application of the proposed method is the remanufacturing of a one-meter diameter cruise ship propeller in aluminum alloy. This component combines several difficulties such as the CAD/CAM data chain to rebuilt, DFAM approach to enable the part to be manufactured by single bead WAAM and part orientation within the working space of the machine.

For this use case, no CAD model was available, and the numerical model was obtained by 3D scanning of a real component. Then the part positioning and orientation into the working space was made, based on the approach presented in Section 2.2. For each blade, the orientation of the substrate was set to limit overhangs and make the manufacturing direction align with the favorable direction. This analysis was made on the CAD software. Then, a DFAM approach was applied on the blades geometry to avoid deposited beads superposition at the leading and trailing edges (Figure 11). The method employed was to modify the CAD model for WAAM operations in order to eliminate the bead covering zones and enable the blades to be manufactured by single bead walls. This modification of the geometry is made possible by the necessity of a finishing operation by machining after WAAM to fulfill the functional requirements on the blade surfaces, especially surface roughness.

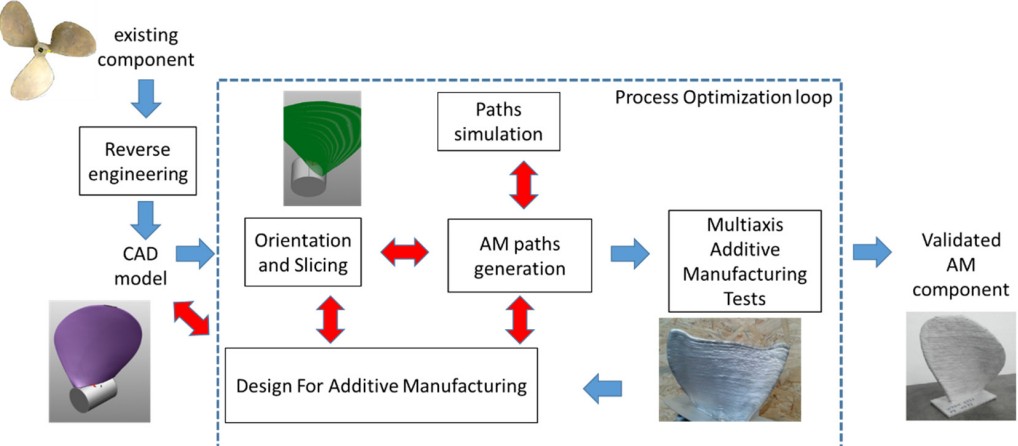

**Figure 11.** Numerical data chain for the propeller use case.

After these single blade manufacturing tests to define the most suitable slicing, blade orientation, and WAAM parameters selection by applying the method given here, the complete propeller could be manufactured successfully (Figure 12).

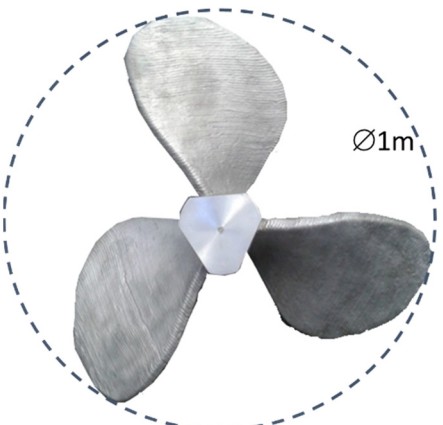

**Figure 12.** One-meter diameter cruise ship propeller made with WAAM in the laboratory.

### 3.2. Aeronautic Structural Panel

This section focuses on the manufacturing of a demo aeronautic structural panel by WAAM. It is a 1000 × 1000 mm fuselage portion as depicted in Figure 13 designed and manufactured thanks to a disruptive and innovative approach: block machined inertia profiles riveted to the fuselage skin have been replaced by additive manufactured stiffeners built directly on the skin. This approach not only simplifies assembly workplans and avoids a series of drilled holes on the structures but also realizes stiffeners geometries very well adapted to the loads paths applied to the structure (due to the ability of WAAM to produce freeform geometries). This saves material also. The buy-to-fly ratios are much better as the material is added only where needed (instead of machining preforms); the mass needed to achieve the same mechanical performances is also reduced compared to legacy approaches as the stiffeners geometry and position can be tailor made by WAAM.

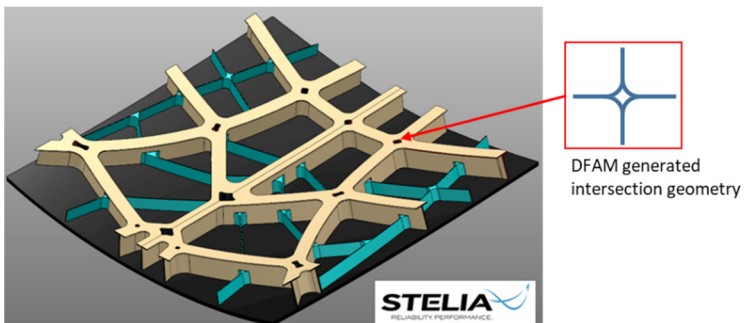

**Figure 13.** Structural panel demonstrator after DFAM.

The demonstrator has a double curvature geometry on which primary ("T" shape, in beige) and secondary (thin wall, in blue) stiffeners are manufactured. The main radius is 2000 mm and the secondary 9500 mm. This demo component is the result of the French DGA/DGAC collaboration project DEFACTO which involved several industrial and academic partners: Stelia aerospace, Constelium, Centrale Nantes, and CT Ingenierie. The objective was to use the new opportunities of WAAM to propose innovative structures for the aircraft of the future.

Compared to usual aircraft structures which are made of profiles assemblies, the concept was to use the external skin as substrate and realize the stiffeners by WAAM. Hence, the process requirement and the design opportunities offered by WAAM have been identified and transferred to the design stage, to generate a geometry well adapted to the manufacturing process, while meeting the functional requirements of any aircraft structure.

In particular, the "roundabout" design strategy has been employed for the primary stiffeners (Figure 13). As the surface finishing process was multiaxis machining, the pattern

at the center of the intersection was adapted to enable a milling tool to machine the external surface of it. This hybrid manufacturing approach that mixes additive manufacturing with high speed machining is another example of Design for Additive Manufacturing enabled with this innovative aircraft fuselage concept.

After the DFAM stage, the component has been manufactured in the laboratory using variable torch axis tool paths to deposit the thin walls (the stiffeners) on the curved substrate, as presented in Figure 14a. Then, the horizontal tail of the primary stiffeners has been realized using the cornice welding strategy. The tail has been considered as two distinctive walls that could be manufacturing independently. A major interest of AM could be employed here with the ability to make wall height and tail geometry vary locally depending on the local loads applied to the structure.

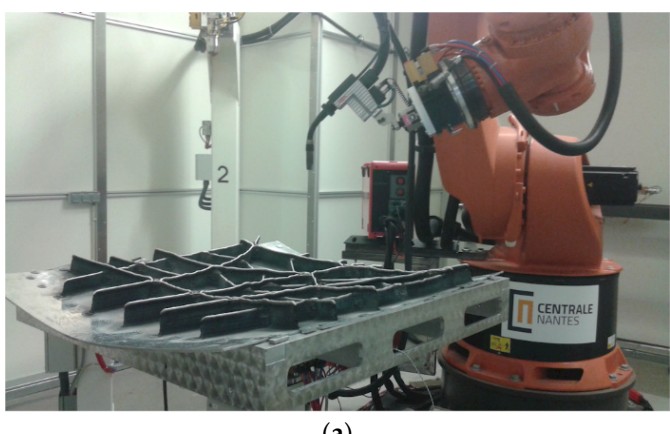 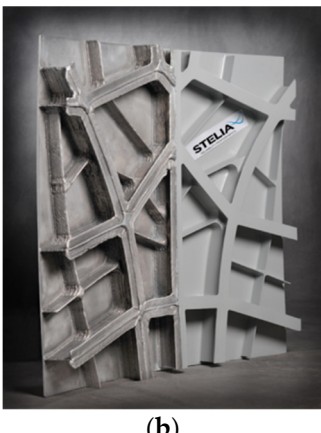

(**a**)  (**b**)

**Figure 14.** Structural panel: (**a**) during the manufacturing process and (**b**) finished.

The resulting part is shown in Figure 14b; the left half has been kept as is after WAAM while the right half has been machined finished and painted as any aircraft structure. The results are very promising for the use of WAAM to realize such structures. Compared to the usual structural components, this demonstrator geometry saves material while better following the load paths on the service. This is made possible by the use of the multiaxis tool paths generation approach for WAAM.

## 4. Discussion

The proposed approach compiles several aspects of tool path generation for the WAAM process. They are integrated in a general methodology that goes from initial CAD model to manufactured WAAM part. For example, the possibility of using non-planar and non-parallel slicing strategies enables the realization of complex geometries by single bead manufacturing. Another singularity lies into the use of a DFAM approach for intersections manufacturing. The main tool path generation strategies to control intersections with WAAM are given in Table 1.

Compared to them, this DFAM based approach has two interests. Firstly, it uses single bead deposition with constant process parameters while being capable of manufacturing any number of tracks at the intersection. Secondly, the CAD model modification applied makes the component ready for surface finishing by milling as sharp corners at the intersections are avoided. This approach is consequently well adapted to hybrid manufacturing workplans because the wall deposited can be machined directly and completely if necessary.

**Table 1.** Main tool path generation strategies for intersections manufacturing with WAAM.

| Research Groups | Tool Path Generation Approach for Intersection Crossing | Reference |
|---|---|---|
| University of Cranfield (UK) | Alternating deposition directions for consecutive layers | [23] |
| | Splitting the geometry in zones and using oscillated tool path systematically | [26] |
| University of Wollongong (AUS) | Replace single bead deposition by 2 or more parallel tool path, based on Medial Axis Transformation. | [27] |
| Southwest Jiaotong University (CN) | Use monitoring to change locally the process parameters and reduce fed material quantity and avoid the apparition of peaks at crossing sections | [25] |
| Centrale Nantes (FR) | Modify locally the CAD model to propose DFAM generated intersections geometries | this paper |

## 5. Conclusions

A tool path generation method for WAAM has been proposed in this paper. Several locks have been addressed, from the numerical data chain control to the tool paths planning based on DFAM considerations. Depending on the characteristics of the component to manufacture, the following rules can apply in particular:

- Rule 1: Multiaxis tool paths are necessary as soon as the component setup in the working space creates material undercuts;
- Rule 2: The deposition direction should be selected after a having determined the favorable direction;
- Rule 3: Using closed tool paths should be privileged to reduce the number of arc starts and stops;
- Rule 4: For open tool paths, alternate travel direction to reduce the effects of arc starts and stops on the deposited beads;
- Rule 5: Cornice welding inspired tool orientation enables to realize horizontal walls without substrate re orientation;
- Rule 6: Use DFAM rules to redesign walls intersections and junctions so that the material deposition is constant.

The use of the proposed method with the associated rules has been discussed and validated by the realization of two large dimensions industrial use cases from the naval and the aeronautic sectors.

**Author Contributions:** Conceptualization, M.R. and J.-Y.H.; methodology, M.R. and J.-Y.H.; validation, M.R. and J.-Y.H.; investigation M.R., J.-Y.H. and V.Q.; data curation, M.R., J.-Y.H. and V.Q.; writing—original draft preparation, M.R. and J.-Y.H.; writing—review and editing, M.R. and J.-Y.H.; project administration, M.R. and J.-Y.H.; funding acquisition, J.-Y.H. All authors have read and agreed to the published version of the manuscript.

**Funding:** This research received no external funding.

**Data Availability Statement:** The data presented in this study are available on request from the corresponding author. The data are not publicly available.

**Acknowledgments:** Some research works of this paper was carried out within the DGA/DGAC funded DEFACTO research project. The partners involved were STELIA Aerospace, CT INGENIERIE, CONSTELLIUM and Centrale Nantes (ECN).

**Conflicts of Interest:** The authors declare no conflict of interest.

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
