# Peer review of "A Multiaxis Tool Path Generation Approach for Thin Wall Structures Made with WAAM"

_jmmp, doi:10.3390/jmmp5040128_

Round 1
Reviewer 1 Report
Dear auhtors,
thank you very much for the interesting work on path planning for WAAM.
The paper is well written and adresses important topics for WAAM like the pathplanning for junctions and tool direction. The presented results for a path planning strategy is concluded within 6 rules which can be easealy adopted.
But i have some comments to improve the quality of your paper:
First the literature survey is not adequate enough. For WAAM there heve been a couple of reviews published in the last yeas wchich adresses path planning and processes. Like http://dx.doi.org/10.3390/app11188619 or https://doi.org/10.1088/1361-6463/ac1e4a and others. Please revise the literature survey and please add and use more findings regarding path planning from the last years.
Second, the paper is missing a discussion section in which you compare your results and strategy with the strategies of other working groups.
Please add a propper discussion section.
Best regards
a Reviewer
Author Response
thank you very much for the interesting work on path planning for WAAM.
The paper is well written and adresses important topics for WAAM like the pathplanning for junctions and tool direction. The presented results for a path planning strategy is concluded within 6 rules which can be easealy adopted.
The authors thanks the reviewers for these comments.
But i have some comments to improve the quality of your paper:
First the literature survey is not adequate enough. For WAAM there heve been a couple of reviews published in the last yeas wchich adresses path planning and processes. Like http://dx.doi.org/10.3390/app11188619 or https://doi.org/10.1088/1361-6463/ac1e4a and others. Please revise the literature survey and please add and use more findings regarding path planning from the last years.
The state of the art has been updated and completed with new papers including these which are suggested.
Second, the paper is missing a discussion section in which you compare your results and strategy with the strategies of other working groups. Please add a propper discussion section.
A discussion section has been added before the conclusion of the paper. A comparison of the main tool path generation strategies for intersections manufacturing with WAAM is proposed and the interest of the proposed DFAM approach is discussed.
Reviewer 2 Report
Reviewers’ Comments
Wire Arc Additive Manufacturing (WAAM) is an important production technique especially for the large components. This article proposes a generic multiaxis tool path generation approach for thin wall structures made with WAAM. It is of certain guiding significance for a methodological reference for understanding the path generation approach of WAAM technology, but the manuscript needs significant improvement before acceptance for publication. My detailed comments are as follows:
(1) The abstract of the article needs to be rewritten and should give some more specific details so as to give more guidance to the reader.
(2) The types and main parameters of experimental equipment should be provided.
(3) What is the meaning of this sentence “An interesting technique, inspired from cornice welding can be employed: Using a predefined tool orientation (for example 20° for aluminum alloys) enables to manufacture horizontal walls such as the example given by Figure 6.” Please give more details.
(4) In Fig.8, the geometry is modified into one central cylinder with four intersecting walls. Each manufacturing layer can be interpreted as a four-tracks roundabout. Is there a theoretical basis for this design approach, relying only on design experience?
Author Response
Wire Arc Additive Manufacturing (WAAM) is an important production technique especially for the large components. This article proposes a generic multiaxis tool path generation approach for thin wall structures made with WAAM. It is of certain guiding significance for a methodological reference for understanding the path generation approach of WAAM technology, but the manuscript needs significant improvement before acceptance for publication. My detailed comments are as follows:
(1) The abstract of the article needs to be rewritten and should give some more specific details so as to give more guidance to the reader.
The abstract has been modified to give more specific details.
(2) The types and main parameters of experimental equipment should be provided.
A paragraph and a figure have been added with some details about the experimental equipment
(3) What is the meaning of this sentence “An interesting technique, inspired from cornice welding can be employed: Using a predefined tool orientation (for example 20° for aluminum alloys) enables to manufacture horizontal walls such as the example given by Figure 6.” Please give more details.
The text and the figure have been modified accordingly to give more details; and the expression “cornice welding” has been replaced by “overhead welding”.
(4) In Fig.8, the geometry is modified into one central cylinder with four intersecting walls. Each manufacturing layer can be interpreted as a four-tracks roundabout. Is there a theoretical basis for this design approach, relying only on design experience?
The text has been modified to explain this geometry has been obtained by design experience with the objective to avoid any point of the intersection to belong to more than two intersecting walls.
Round 2
Reviewer 1 Report
Dear authors,
i have no further comments, all requested changes have been made.
Best Regadrs
a Reviewer
Reviewer 2 Report
The author responded very well to the reviewer’s questions, and the manuscript has been further improved and can be accepted.